# Data Compression in the NEXT-100 Data Acquisition System

**DOI:** 10.3390/s22145197

**Published:** 2022-07-12

**Authors:** Raúl Esteve Bosch, Jorge Rodríguez Ponce, Ander Simón Estévez, José María Benlloch Rodríguez, Vicente Herrero Bosch, José Francisco Toledo Alarcón

**Affiliations:** 1Instituto de Instrumentación para Imagen Molecular (I3M), Centro Mixto CSIC, Universitat Politècnica de València, Camino de Vera s/n, 46022 Valencia, Spain; rponcejorge@gmail.com (J.R.P.); viherbos@eln.upv.es (V.H.B.); jtoledo@eln.upv.es (J.F.T.A.); 2Nuclear Engineering Unit, Faculty of Engineering Sciences, Ben-Gurion University of the Negev, P.O. Box 653, Beer Sheva 8410501, Israel; ander@post.bgu.ac.il; 3Enrico Fermi Institute, University of Chicago, Chicago, IL 60637, USA; 4Donostia International Physics Center (DIPC), Paseo Manuel Lardizabal 4, 20018 Donostia-San Sebastian, Spain; jmbenlloch@dipc.org

**Keywords:** xenon TPC, data acquisition circuits, FPGA, data compression techniques

## Abstract

NEXT collaboration detectors are based on energy measured by an array of photomultipliers (PMT) and topological event filtering based on an array of silicon photomultipliers (SiPMs). The readout of the PMT sensors for low-frequency noise effects and detector safety issues requires a grounded cathode connection that makes the readout AC-couple with variations in the signal baseline. Strict detector requirements of energy resolution better than 1% FWHM require a precise baseline reconstruction that is performed offline for data analysis and detector performance characterization. Baseline variations make it inefficient to apply traditional lossy data compression techniques, such as zero-suppression, that help to minimize data throughput and, therefore, the dead time of the system. However, for the readout of the SiPM sensors with less demanding requirements in terms of accuracy, a traditional zero-suppression is currently applied with a configuration that allows for a compression ratio of around 71%. The third stage in the NEXT detectors program, the NEXT-100 detector, is a 100 kg detector that instruments approximately five times more PMT sensors and twice the number of SiPM sensors than its predecessor, the NEXT-White detector, putting more pressure in the DAQ throughput, expected to be over 900 MB/s with the current configuration, which will worsen the dead time of the acquisition data system. This paper describes the data compression techniques applied to the sensor data in the NEXT-100 detector, which reduces data throughput and minimizes dead time while maintaining the event rate to the level of its predecessor, around 50 Hz.

## 1. Introduction

### 1.1. Introduction to NEXT Detectors

The NEXT-100 detector [1,2] is the third phase of the NEXT detector series and is expected to operate by the end of 2022 at the Laboratorio Subterraneo de Canfranc (LSC) in Spain. The experimental goal of NEXT experiments is to search for neutrinoless double beta decay in ^136^Xe using high-pressure xenon gas time projection chambers (HPGXeTPC) with amplification of the ionization signal by electroluminescence (EL), which offers good energy resolution and tracking-based event identification. Moreover, the NEXT collaboration is currently defining a ton-scale version of NEXT-100 [3] that would be able to reach a sensitivity to the half-life of the ^136^Xe neutrinoless double beta decay of 10^27^ yr, after a few years of operation.

In the NEXT detectors built up to the present time, the interaction of charged particles with xenon gas is immediately followed by the emission of scintillation light, the so-called primary scintillation (S1) signal. The ionization electrons left behind by the interacting particle drift under the influence of an electric field toward another region of the detector, the EL gap, with an electric field of higher strength. There, the electroluminescence light (S2) signal is emitted isotropically, with intensity proportional to the number of ionization electrons. Until now, S1 and S2 signals have been detected by two different types of photosensors installed on opposite planes: the Energy Plane (EP), based on photomultipliers (PMT) for precise energy measurement, and the Tracking Plane (TP), based on a dense array of silicon photomultipliers (SiPM) for topological event filtering.

### 1.2. Motivation

Searches for neutrinoless double beta decay require excellent energy resolution to eliminate background events that occur at energies similar to the Q-value of decay. The energy response is characterized by means of calibration sources to yield energy peaks over a range of energies from several tens of keV up to and including Q_ββ_, that is 2.458 MeV. Alpha, electron, gamma, xenon characteristic X-ray events, and muons must be studied to understand the detector in a wide range of energies, as well as to measure its energy resolution [4,5,6,7]. The calibration process will be one of the major challenges in NEXT-100 since it implies reading out sensor data at the maximum possible rate allowed by the DAQ system.

The calibration process is well known thanks to the NEXT-White detector [8], a 5 kg prototype of NEXT-100 at 10 bar that has been operated from the end of 2016 to June of 2021 at LSC. NEXT-White was instrumented with 12 PMTs (Hamamatsu R11410-10) in the EP and 1792 SiPMs (SensL series-C) in an array at a pitch of 10 mm in the TP. NEXT-White was continuously calibrated by pumping ^83m^Kr into the system. The point-like nature of such events, given their low energy (41.5 keV), makes them ideal for determining the electron lifetime and energy response throughout the detector [6]. For NEXT-White, it was deemed necessary to use at the very least 6.5·10^5^ events to properly characterize the detector in a 24 h interval. The overall selection efficiency for such events was measured to be 65.5%, which imposes an acquisition requirement of ~11.5 Hz (around 1 million events per day) with a data buffer set to 1.6 ms.

Nevertheless, as can be seen in Figure 1, NEXT-100 is a 100 kg detector at 15 bar instrumented with 60 of the same PMT device used in NEXT-White and 3584 SiPM (Hamamatsu S13372-01) in an array at a pitch of 15.55 mm. Moreover, NEXT-100 will have a ~1.2 m drift region. For a 400 V/cm drift field, similar to that of NEXT-White, and an operational pressure of 15 bar, the maximum drift time is expected to be approximately 1400 µs, considering a drift velocity of 0.835 mm/µs (obtained through Magboltz simulations). Therefore, a maximum buffer size of 3.2 ms, around 2.3 times larger than the maximum drift, should easily accommodate all types of events, including muons fully traversing the detector along the longitudinal dimension.

Given the increase in the number of sensors, detector volume, and data buffer size, it is obvious that reading out events in the NEXT-100 poses a stronger requirement to the acquisition system. Simulation studies with different lifetimes and organic wavelength shifter tetraphenyl butadiene (TPB) imperfections have been carried out within the NEXT collaboration to assess the specifications of the new detector. The simulations have yielded a requirement of at least 3·10^6^ events per day in a realistic scenario (10 ms electron lifetime, imperfections similar to the ones observed in NEXT-White) and up to 5·10^6^ events in a pessimistic scenario (lifetime would vary up to 1 ms) during a 24-h period and a minimum buffer size of 1.8 ms, in low energy calibration mode. Although these numbers are expected to be mitigated by improvements in the calibration procedure, they remain the target for calibration procedures. This implies that the acquisition event rate will increase by a factor between 4.6 and 7.6 (event rate in the range from 50 to 90 Hz) and with a larger data buffer.

The Data Acquisition System (DAQ) and the Event Detection System (EDS) [9] in NEXT-100 have been properly scaled to the new detector requirements in terms of the number of sensors and the data buffer. Nevertheless, the detector will pose a challenge to the DAQ due to the expected increase in the number of events and their size produced in the detector, as stated above. The DAQ is organized into modules where data must be read out from a buffer and sent to a server through two Gigabit Ethernet (GbE) links per module with limited bandwidth, which implies a clear bottleneck in every DAQ system. In NEXT-100, seven servers should handle the event data load estimated from 920 to 1520 MB/s. These values are obtained extrapolating the throughput per sensor of NEXT-White to NEXT-100, assuming the TP data are already zero-suppressed with a compression ratio of about 70%, with the maximum buffer size of 3.2 ms (worst case) and a rate being from 4.6 to 7.6 times higher. In this case, approximately 68% of the data corresponds to the EP data sensors and the rest to the TP data sensors. Nevertheless, tests show that each server is able to read out and store data on disk at a maximum rate of 125 MB/s that gives the system the possibility to reach a maximum throughput of 875 MB/s, that is below the needs of the calibration procedure.

This problem can be directly solved by increasing the time needed for calibration, which implies reducing the calibration rate or scaling up the DAQ, accommodating sensor data in more servers. However, in addition to enriching the DAQ and the EDS with a set of features thought to reduce the data throughput, it is clear that the DAQ could implement additional features that help the system accomplish the calibration process in a reasonable amount of time, such as improving the current data compression.

### 1.3. Considerations

In the case of the NEXT-100 DAQ, the main goal of the data compression algorithm that could be applied is to reach a very high rate of compression to reduce data throughput with minimal hardware resources.

To accomplish this goal, some considerations must be taken into account regarding the compression module needed:**Limited hardware resources.**The available DAQ resources are limited since the algorithm must be implemented using existing FPGA devices that are already used to read out detector data. The compression module must be implemented with minimum hardware resources. On the one hand, this implies keeping the algorithm as simple as possible. On the other hand, it is desirable to have the algorithm uncorrelated from the number of sensors to be processed, avoiding parallelizing the module or parts of it, if possible. It is important to remark that the detector, at least in the TP, has a very large number of sensors to read out per DAQ Module (up to 768 sensors). Related to this, module placement in the data chain for both sensor planes could have a considerable impact on the hardware resources needed, so this must also be carefully studied.**High compression ratio.**Different algorithms accomplishing statement one must be studied to select the best option with a minimum, but high, compression ratio. As stated in Section 1.2, the event data load estimated due to the calibration process will be in the range of 920 to 1520 MB/s, while the maximum system throughput will be about 875 MB/s. A minimum reduction factor of 2 is needed, but a better compression ratio will help to reduce dead time since its value is related to the data acquisition system throughput.

## 2. NEXT-100 Data Acquisition and Event Detection Systems

### 2.1. Hardware Architecture

As mentioned above, the NEXT-100 DAQ and EDS are scaled versions of the previously implemented NEXT-White detector. The hardware is based on the SRS-ATCA (Scalable Readout System ported to the Advanced Telecommunications Computing Architecture standard) jointly developed by the NEXT Collaboration, CERN-PH and IFIN-HH Bucarest in the framework of the CERN RD51 collaboration [10,11]. The main module, the SRS module, based on FPGA, provides a customizable interface with a set of generic plug-in cards. As an online system, the DAQ will use DATE (ALICE Data Acquisition and Test Environment) [12].

The SRS DAQ module provides real-time digital processing through two Xilinx Virtex-6 FPGAs (XC6VLX240T-1ff1156). Each FPGA is connected to a DDR3 SO-DIMM memory module, which is used as a double-data memory buffer. Two on-board custom mezzanine connectors provide I/O flexibility for a wide range of front ends. Each group of FPGA, memory and mezzanine connectors can be used as an independent processor unit. In NEXT detectors, two different mezzanines are used: the EAD-M1 unit (ADC Card), with 24 ADC channels (12 bit, at 40 MHz) is the interface with the PMT sensors front-end; and the DTC (Data, Timing and Control) card, with 12 DTC links [13] on HDMI connectors, is used to interface the Front-End Board (FEB) [14] at a maximum speed of 200 Mb/s. FEBs provide the analogue front-end electronics for 64 SiPM sensors, analogue to digital conversion at 1 MHz (12 bit), and digital processing through a Xilinx Virtex-6 FPGA (XC6VLX130T-1ff784). DTC links are used for configuration, data transfer and trigger purposes. The SRS DAQ module also includes a connection to a Rear Transition Module (RTM) for several GbE connections and other I/O connectivity as two HDMI (High Definition Multimedia Interface) for DTC connection and external input/output NIM (Nuclear Instrumentation Module) connections for external trigger purposes. In NEXT-100, 4 GbE connections are used, giving a maximum throughput per blade of 500 MB/s.

### 2.2. System Architecture

As shown in Figure 2, seven SRS DAQ modules (in the figure “ATCA blades”) are needed to read out the data from the EP and TP planes, as well as to perform event detection and system control. Each SRS DAQ module independent processor unit is devoted to read out PMT sensors (up to 12 PMTs per module, up to 84 in total) or FEBs (up to 12 FEB boards per module, 72 in total). The system has a total of 13 DAQ Modules (divided into 7 EP DAQ Modules and 6 TP DAQ Modules), and for control and event detection duties, one extra Control Module. The connection between modules and FEBs is given by the DTC links.

As seen in Figure 3, the readout works in push mode. DAQ Modules read out, time stamp and store data coming from the front end in a reconfigurable-length circular buffer, whose maximum size corresponds to approximately twice the maximum detector drift time (up to 3.2 ms). The circular buffer is indeed a double circular buffer implemented on the DDR3 memory. Data are sent from the DAQ Modules to the servers using two optical 1 Gb/E links per module. Each server is able to read out and store data on a disk at a maximum speed of 125 MB/s. DAQ modules are connected to the servers in such a way that the data load is equalized between them.

In NEXT-100, the Event Detection System (EDS) is based on the early energy measured in the PMT sensor. At this stage, event candidates are generated, which are sent to the Control Module, where a processor generates an Event Accept signal that produces a data upload from each DAQ Module buffer to the online system for later offline analysis.

The EDS is based on a two-processor architecture. Each pair of processors can be configured with a set of parameters to search for a different type of event. The most obvious application of these double-event detection features is to allow calibrations to be carried out while taking physics data, ensuring high-quality and properly calibrated physics data. Moreover, the use of a double circular buffer can be associated with the type of event, allowing different modes of operation. For instance, one of those modes gives priority in the use of the double buffer to a type of event, generally devoted to detecting physics data since it guarantees the minimum dead time for this type of event.

A compression module (zero-suppression algorithm) has already been implemented in FEBs. However, since each DAQ module is based on an FPGA device, additional compression techniques can be added before sending data to the servers to reduce data throughput.

## 3. Data Compression in Physics Experiments

Real-time data lossless compression algorithms implemented on hardware are shyly present, or at least well documented, in data acquisition systems for high-energy physics experiments conceived for pulse digitizers. One of the earliest references can be found in the implemented lossless compression for the electromagnetic calorimeter in the CMS experiment [15]. The implemented module shows how Huffman encoding applied to a bounded set of codes, due to its simplicity, can be used to significantly reduce data throughput. A similar scheme has been implemented after zero suppression, with a very short range of Huffman code, in the MicroBooNE TPC [16].

Similar schemes, with more complex signal pre-processing, combined with zero suppression, have been used in different detectors, such as PANDAX-III [17] or ALICE TPC [18], with very high compression ratios. In all cases, the limited available hardware resources pose a design constraint.

A deep study of the signal characteristics and the suitability of different approaches to the detector’s signal is of paramount importance to reach the maximum compression ratio with minimal hardware resources. The compression modules described in this article are based on the ones reported in the above references, although they reach similar results with simpler implementations, avoiding signal pre-processing and the introduction of noise.

This paper presents a study of different techniques for lossless compression, as well as alternatives in the codification process, with applications to digitized signal pulse waveforms. The application is then extended with excellent results to the digitized integrated charge of SiPM sensors, combining zero suppression and a lossless compression algorithm.

Another interesting contribution is the study of the dynamic reconfiguration of Huffman codes. Although feasible, we reported a low impact on the compression ratio.

## 4. Data Compression Techniques

### 4.1. Introduction

As stated in the introduction, data compression can be useful to reduce the limiting throughput of the DAQ, at least if the compression, data transmission and decompression procedures are faster.

As is well known, in a basic way, data compression can be divided into two main classes: lossless compression and lossy compression. In the first case, it is possible to obtain, from a compressed message, the complete original message, while with the second, it is only possible to obtain a fraction of the original message. It is not acceptable in all cases to lose data since it can damage the integrity of the message. Nevertheless, in some cases, it is possible to reduce the amount of data to be sent by rejecting irrelevant information, and this is justified by the fact that these techniques can provide higher compression ratios than lossless methods.

In addition to this classification, it is possible to distinguish the adaptability of a compression method. A non-adaptive mechanism is rigid and does not allow the modification of any of its operating parameters, while an adaptive method performs a process of examination and modification of the parameters according to the compressed data.

There are many data compression techniques of both types, but not all of them are applicable to DAQ systems since most of them are based on FPGA or other types of processors that are limited in speed and hardware resources.

The following sections describe the algorithms that have been analyzed in the context of NEXT experiments.

### 4.2. Lossy Compression Techniques

In particle detectors, in addition to the event detection system (trigger), which filters adequately the physic interesting events, helping to reduce the detector throughput by selecting events of interest, the most common compression method implemented is zero suppression (ZS). This type of compression allows data to be sent above a configurable threshold over the baseline signal, including as optional a set of pre- and post-samples. In this case, extra bits are needed since data must be time stamped and channels flagged, but the data reduction reached compensates for it. Moreover, it does not need excessive digital hardware resources and the data reduction ratio can vary depending on the configuration parameters set.

Other lossy compression algorithms applied to other types of data, such as pictures, video or music, are out of the scope of this study since they are too complex to be integrated in the detector readout chain and, anyway, they do not guarantee the relevant physics data content.

### 4.3. Lossless Compression Techniques

There are compressors that use the statistical characteristics of the source to obtain optimal encoding. These compressors are called statistical compressors. They start with a finite number of messages whose different possibilities are known, either experimentally or fixed. Its goal is to encode the different messages from the source in such a way that the resulting data have fewer bits than the original. A good example of this type is Huffman coding [19]. This algorithm is based on a tree of codes of different sizes that are assigned to the possible messages of the data as a function of the probability of appearance. This implies that the most likely message has the shortest encoding.

There are algorithms, such as substitutional algorithms, that are based on a dictionary with strings of different messages. Each string has an index, and if the message has already been sent before, only the index preceding that string in the dictionary is sent.

Another example of lossless compression is Run Length Encoding (RLE) [20], where sequences of the same data present in the series are encoded as a single value and a counter indicates the repetitions of consecutive values. For physics data, it will provide efficient compression if the same sequence of values appears repeatedly and consecutively in the signal.

Other lossless compression algorithms, such as Context Tree Weighting (CTW) [21] and LZ77/LZ78 [22], are more complex to implement and are not efficient with physics data, as they do not have the same characteristics as text data.

### 4.4. Signal Conditioning

Additionally, when the signal to be compressed presents smooth transitions, delta encoding can be applied prior to compression. This technique modifies data as the difference between successive values, thereby reducing the variance of the values. The first value corresponds to the original, while the subsequent values in the data stream present an increase or decrease with respect to the previous one.

Another way to obtain similar results is to subtract the DC value (mean value of the signal), also called the baseline, which is defined as the electrical signal from a sensor when no measured variable is present. All the values sent are the difference with respect to the signal baseline. In this case, to recover the signal, the extracted mean must be sent to the stream.

Both cases are very useful in many data series, where there is a certain correlation between the data, oscillating around an intermediate value. Encoding sets of values that have a high mean value can lead to the excessive use of bits for storage. However, they may present differences between their maximum and minimum values that are much lower than the indicated continuous level.

In addition to the advantages stated above, in some cases, as happens in NEXT, where the number of channels to process is high, these techniques help to equalize the signals of different channels. This is helpful when some compression algorithms are applied, such as those based on tables with codes to be replaced since it allows the use of a unique set of codes instead of one per signal to be processed.

## 5. NEXT Experiment Data Compression Study

### 5.1. Lossy Data Compression Review

In NEXT experiments, as in many other experiments, the sensor data acquired in a certain window can be divided into two parts: a pulse preceded and followed by a stable signal and the baseline of the signal, as it has been already mentioned, in a certain window. In the NEXT experiments, the absence of a pulse clearly dominates the signal, as seen in Figure 4.

All NEXT detectors apply zero-suppression as a compression algorithm. ZS parameters need to be chosen carefully to avoid biasing events, as the charge loss is minimal. With particular interest in the energy and temporal position of the pulses, this information is preserved by keeping the signal pulse and sufficient samples, before and after the pulse, to estimate the baseline of the signal.

The algorithm applied has a set of configurable parameters: a threshold over the baseline to consider a pulse, the minimum number of consecutive time bins that have to be above the threshold to consider a pulse, and a set of pre- and post-samples to be sent together with the considered pulse.

A study performed with experimental data shows that a high reduction factor can be reached with a very low charge loss, as can be seen in Figure 5. It must be noted that for NEXT-White off-line data processing, the SiPM signals are cut much stronger than the ZS threshold employed at the DAQ level. Specifically, for ^83m^Kr events, a minimum of 5 photoelectrons (corresponding to ~75 ADCs) is required per pulse and sensor. This was deemed to be more than enough to achieve good precision for point-like events. On the other hand, for high-energy events, the threshold is increased to 10 photoelectrons per 1 µs time slice. This value was identified as the optimal threshold for track reconstruction [23]. Still, given the non-reversibility nature of lossy compression, a heavily conservative ZS configuration was chosen for standard data-taking. In fact, the region slightly above 75% reduction was identified as ideal, as charge loss was negligible while maintaining a strong reduction factor.

In the case of EP, no lossy compression algorithms can be applied. On the one hand, any charge loss produced by a ZS applied would have a negative impact on the detector energy resolution requirement (less than 1% FWHM). On the other hand, due to the capacitive coupling of the PMT signal [24], the baseline needs to be restored by a baseline restoration (BLR) algorithm. The effect of these DC-rejecting capacitors on the obtained analogue signals is similar to that of a high-pass filter with a very low cutoff frequency. Due to the high energy resolution requirement of the detector, it has been decided to send data in raw mode and compensate afterwards by software, allowing a precise reconstruction of the original PMT output signal with a negligible error. This affects the size of the pulses, as can be seen in Figure 6, which are longer than usual. In addition to the energy resolution issue, this effect on the signal would have a negative impact on the compression ratio that could be reached by a ZS algorithm.

### 5.2. Lossless Data Compression Study

The histograms in Figure 7 clearly show that there are codes more probable than others, being the signal suitable for statistical compressors, such as the Huffman algorithm or other dictionary-based techniques. As can be seen in the histograms, the maximum code frequency corresponds to the sensors DC value, which is set around 2300 and 50 for PMT and SiPM sensors, respectively. As is appreciable in Figure 7a, not all sensor signals are set on the same baseline.

The different algorithms proposed in this article are based on the idea presented in [15], in which only a set of possible codes is used by the compression algorithm. On the one hand, this implies finding the best set of codes and its quantity, which simplifies the compression module while keeping an optimum compression ratio. On the other hand, this forces the compression module to set a way to differentiate encoded codes from the rest.

Under those assumptions, several algorithms have been studied using Python with numpy and scipy libraries. In the simulations, a flag is used to differentiate encoded and non-encoded data. The options studied are divided into two levels as a function of how data are prepared prior to compression and, inside each first level, how data are encoded. The first level is based on:**Baseline subtraction**. In this case, the DC value of the signal is subtracted prior to compression.**Delta encoding**. In this case, delta encoding is applied prior to compression.

The second level is based on:**Ca2**. Data in the compression range are encoded by its two’s complement values with the minimum number of bits needed. This codification is direct and very simple.**Sensor ref + Ca2**. Prior to encoding the data, reference sensor data (sent without compression) is subtracted from the rest of the sensor data. Then, case 1 is applied.**RLE**. The data are RLE encoded.**Huffman**. Data in the compression range are Huffman encoded. It requires the calculation of the Huffman tree to set the codes and their sizes in bits.

### 5.3. Conclusions

Both types of signal conditioning, delta encoding and baseline subtraction-based, can be used with PMT sensor data. This is illustrated in Figure 8, which shows the differences in performance for various combinations of signal conditioning and data compression algorithms. However, delta encoding offers better results in general when the same type of compression encoding is applied.

In both cases, coding the data with a lower number of bits—in this case Ca2 or RLE—has a peak of around 4 to 8 codes (2 to 3 bits) and a maximum compression ratio of 60 to 70%, decreasing considerably if more codes are used for compression. Those algorithms work fine with a short number of codes but have a clear limit in compression ratio of around 70%.

In the case of Huffman encoding, unlike the previously mentioned algorithms, the compression ratio increases as the number of codes applied increases and remains quite stable over 8 codes. The reason is that each sample is compressed with a code and a number of bits established by the Huffman tree, which is lower depending on the probability of appearance and not with the same established number.

In the case of the TP, algorithms based on a signal reference produce very bad results. The number of channels discourages the use of algorithms based on Baseline Subtraction due to the high number of hardware resources needed due to the larger number of sensors to process per DAQ Module. From the set of algorithms based on delta encoding, the best results are obtained when Huffman codes are used, as seen in Figure 9. The behavior of algorithms based on RLE compression or the use of other types of codifications, such as Ca2, draws similar results to the EP case.

Subtracting the DC value from the signal sensors implies more hardware since the DC value of each sensor signal must be calculated, needing an extra module per channel (a moving average filter, for instance), and the limited precision obtained can considerably affect the efficiency of the compression algorithm and have, in general, the worst results for the same compression algorithm. In the same way, having a sensor signal as a reference leads to worse results, and it is discarded. For a delta encoder, extra hardware is also needed, but a set of registers or a memory, some control logic and just one adder is sufficient for any number of sensor data to be processed. Compressing the data stream using Ca2 codification responds to the necessity of having a very simple algorithm. Once the signal is conditioned, the values are already Ca2 encoded and only have to be shortened by a number of bits, but the maximum compression ratio, as it happens with RLE, is below the one obtained with Huffman encoding. Therefore, the simplest algorithm to implement, in addition to being the one that yields higher compression ratios, in both cases, with a minimum number of codes, corresponds to Huffman coding with a delta encoding, as in the previous stage. In this case, as seen in Figure 8 and Figure 9, a set of codes of 16 to 32 seems to be optimal since the compression ratio reached is very high without the need for high memory resources.

## 6. NEXT Experiments Huffman Coding Implementation

### 6.1. Control Codification

As stated in Section 5.3, delta encoding prior to Huffman compression seems to be the best option for NEXT-100 sensor signals. Moreover, since a set of Huffman codes is used and not all the possible data values are encoded, some type of control coding is needed to distinguish Huffman encoded data from those that are not. Two possible methods have been studied:**Method 1**. An additional control code of one bit is used to distinguish between encoded and non-encoded data.**Method 2**. Only non-encoded data are flagged. In this case, a non-used Huffman code can be used prior to sending the non-encoded data.

For both types of sensors, the compression procedure can be described as shown in Figure 10. In the figure, input data are supposed to already be delta encoded. In addition to codes that are out of the selected code range, the first event data for each sensor are sent without being encoded. A sensor mask defines the sensors whose data are present in the frame (not shown in the figure). In raw mode, the sensor mask is sent only once. Data are preceded by a timer (16-bit Fine Timer, FT) that defines its position in the DAQ circular data buffer. Data are always sent starting with the sensor with a lower index. Sensors are identified by their positions in the data frame. In the case of SiPM sensors, the FEB ID is also available in the data stream, so the sensors are fully identified by their position in the data frame and the mentioned ID. In the case of Method 1, control codes are inserted before the data. In the case of Method 2, control codes are inserted only in the case of non-encoded data. The number of bits needed in the latter case depends on the number of Huffman codes used since the control code should be the first Huffman code available and not used in the set of selected codes.

Figure 11 shows a comparison of both control encoding methods using up to 32 Huffman codes. As can be seen, when the probability of the appearance of the codes selected for Huffman coding increases, Method 2 tends to have less overhead than Method 1.

Since the data format base word size is 16 bits, if the group control word plus data does not exceed this number of bits, the compression module is easier to implement. For this reason, a maximum of 4 bits control code has also been studied as a particular case of Method 2.

Table 1 shows the compression ratios for the proposed control codification methods. Method 2 has a 5% to 6% better compression ratio for both types of sensor signals. However, the use of a 4-bit control code with Method 2 (Method 2b) yields similar results and simplifies the implemented circuit. However, one of the objectives of the compression module is to compress already zero suppressed data, and in this special case, the compression ratio of the simulated methods is similar.

Considering the facts stated above, and in order to have a uniform control codification method that simplifies the algorithm implementation and decoding process, Method 2b has been chosen.

### 6.2. Huffman Encoding Implementation

As stated in Section 1.3, the compression module has been designed considering the use of minimum hardware resources. This module is divided into two blocks: Delta Encoder and Huffman Coding Module. While the first needs to be sensor related, the second is applied to the data chain independently of the sensor being encoded. This can be done when all data sensors are serially read and the sensor number can be identified. As shown in Figure 3, this can be done, in the case of the EP DAQ Module, at the output of the module, prior to format and sending the data to the online system, and for the TP DAQ Module, at the input where data from different FEC cards are readout in round robin, and re-format prior to storing them in the event circular buffer.

Figure 12 shows the data block of the Compression Module. As shown, to perform delta encoding for each sensor, one data sample must be stored. Control of the sensor processed is made by a control module (Format Control). The stored data samples are subtracted from the new arrival samples. Delta-encoded data are compared with the range of symbols that will be compressed. A control module (Data Encoder Control) takes care of the codes to be Huffman encoded, the size of the data, the insertion of control codes, and to restore the data format with other control codes, such as timing information, card identification and sensor mask, and padding bits if needed. Huffman codes and their sizes are stored in 2 register banks and can be configured through commands. Although some latency is added to the compression process, the compression itself reduces the dead time due to the reduction of data to be sent through the available links.

Table 2 shows the amount of hardware resources needed for the Compression Module. As shown, the amount of resources is reduced. In the case of the TP, more resources are needed. This is because, on the one hand, Delta Encoder Memory is implemented with RAM resources instead of Slice Registers. In this case, storage is needed for up to 768 12-bit sensor data words. On the other hand, managing the data stream with a higher number of sensors from different modules and with different format types (raw or zero-suppressed data format), as described in Section 6.3, makes the Delta Encoder more complex. In the case of the EP since the module is placed at the output, one module per data link is implemented, with two modules in total.

In the case of the TP sensors, data compression is distributed among modules. Zero-suppression is performed in the FEBs and lossless compression is done upon the arrival data signals in the TP DAQ Modules. This allows us to redistribute the hardware resources needed to implement both types of compression.

As shown in Figure 8 and Figure 9, it was decided to use a set of 16 and 32 Huffman codes for the EP and TP sensors, respectively.

### 6.3. Huffman Encoding with Zero-Suppressed Data

In the case of the TP, different scenarios can be applied. As mentioned in Section 5.1, zero suppression has been applied in all NEXT detectors. Including an extra module of lossless compression allows for a configurable compression scheme: zero-suppression, Huffman encoding, or both.

In the case of zero-suppressed data, the data stream has some constraints that must be considered. First, every timing-related data must be preceded by a set of control words indicating time (FT) and data sensor present (MASK) since sensor data can be present or not in the data stream for each time bin. Second, the data stream presents groups of data that are not continuous in time defined by the configured pre- and post-samples, as shown in Figure 5a, which can be an issue for the compression module.

To avoid data overhead and simplify the Delta Encoder Module, it was decided to process the data stream in a continuous mode. As can be seen in Figure 13, each time a block of zero-suppressed data is compressed, the starting data word of the block is considered as if it were the next word of the previous one. In this way, the procedure is exactly the same as that described in Section 6.1.

Figure 14 shows that for 32 Huffman codes, the compression ratio that can be reached by applying only a lossless compression method is around 60%. This is significantly lower than the achieved application of only zero-suppression with a conservative configuration that is around 71%. Merging both types of compression, the system has an extra reduction factor of around 8% and around 79% of compression in total. In this test, different Huffman encoding tree have been applied for Lossless compression and ZS + Lossless compression. With the same Huffman trees applied to both cases (the one obtained for non-ZS data), the compression ratio differs by 0.00015%. This means that even though the Huffman trees obtained slightly different, ZS does not have a significant impact on the code frequency.

### 6.4. Dynamic Versus Static Reconfiguration

Huffman codes require knowledge of the probabilities of the codes in a data stream prior to obtaining the corresponding Huffman tree with the codes and its sizes that will be used to encode the data stream. The codes can be obtained offline, studying events of different previously taken sources. This mechanism minimizes the use of hardware resources since only the obtained codes and the number of bits of each code must be stored in a memory and can be easily configured through commands.

A study of different types of physics events and the number of events to consider to get the Huffman tree has been conducted. As can be seen in Figure 15, apart from a few of the selected codes, the rest have a probability of appearing in the data stream below 1%. Only 5 or 13 codes are over this threshold in the case of the EP or TP, respectively. This fact reinforces the idea of having a limited set of codes of 16 and 32 depending on the detector plane.

Moreover, as shown in Figure 16a, the compression ratio variance when different numbers of events are used to set the Huffman tree is minimal. The same happens when different types of data sources are studied, as seen in Figure 16b. In this case, for 300 events and the special case of using only one sensor as a reference instead of the whole set of sensor data, the difference is below 0.02%. The number of events used, or the type of run, to set the Huffman tree results in a different set of codes. However, the number of bits of the associated Huffman codes remains very stable in all cases and the codes involved as well. In any case, if there is a change in some of the Huffman codes, it happens in those with less probability of appearance, which would imply a minimum change in the Huffman tree, as in the final probability ratio.

With a similar study, but for the case of the TP sensors, similar results are obtained.

These studies justify the use of configurable but fixed Huffman codes instead of a more complex module that integrates a dynamic reconfiguration since the impact on the compression ratio is under 0.05%.

However, since NEXT-100 data are not available, a module to dynamically obtain the codes has been implemented in the PMT module, on the one hand, to understand its implementation and, on the other hand, to understand the need for hardware resources. This module must work in parallel to calculate the Huffman tree in real time. As shown in Figure 17, the module is divided into 4 main blocks:

Statistics Module.

This module calculates the histograms for a selected channel over a fixed range of possible values. Doing it for all the channels and the complete range of possible codes (4096) would consume a large amount of hardware resources. Histograms are calculated for a set of fixed values that cover twice the set of codes needed.

2.Tree Module.

This module generates the Huffman tree. This module selects the most probable codes and assigns the corresponding Huffman codes. This procedure is iterative.

3.Codification Module.

This module stores the new Huffman codes and sizes in their memories and sends the codes to the software decoder placed on a server.

4.Dynamic Reconfiguration Control.

This module takes care of the number of events to be processed to set a new set of codes (a reference number of around 300 events, as seen in Figure 16a, can be used). It also controls the procedure. Histograms are continuously being calculated, but codes are defined if code probabilities change or every certain amount of fixed time.

Table 3 shows that the amount of resources needed is affordable. However, resource utilization in the DAQ Modules will be incremented substantially, especially in terms of occupied slices, making the mapping and routing process more difficult. For this reason, together with the fact that, in the case of having a set of fixed Huffman codes, the impact on the compression ratio stays lower than 1%, has led to the decision not to include the module in this phase of the detector. Since this module is implemented and available, it will be considered in future versions of the acquisition system.

### 6.5. Decoding Software

The compressed data for every event are sent by the FPGA to a computing farm where the DATE-based online system is running. This information is stored in binary files with a specific format defined by the NEXT collaboration. For each file, decoding software is run to convert them to a higher-level format, HDF5 [25], which is used by the reconstruction and analysis software developed by the collaboration.

To successfully implement a compression mechanism in the DAQ, the FGPA must be able to compress the data and the decoding software to decompress it. This software has been updated to read the Huffman-encoded information. To achieve this, the Huffman tree for each type of sensor (SiPM and PMT) has been added to a MySQL database. This information is indexed by the run number so it can be changed over time easily. The decoding software reads the trees from the database once again.

The memory footprint of this update is negligible since it only requires two small Huffman trees in memory and the latest read value for each sensor. The processing time can be longer than in the uncompressed case since the algorithm is more complex, but it should not represent a significant problem.

## 7. Results

The compression modules presented have been integrated into the data acquisition chain and have verified their functionality with real data.

In the case of the EP sensors, the module was ready before the shutdown of NEXT-White, and some runs were taken to study its performance. The detector is equipped with 12 PMT sensors. Table 4 summarizes the results obtained. As can be observed, the compression ratio is 82.3%, a similar value to that obtained in the simulation.

In the case of SiPM sensors, the implemented module has been validated in the NEXT-DEMO++ detector. This detector is a smaller version of NEXT-100 (equipped with 256 SiPM) installed at IFIC (Instituto de Física Corpuscular, Valencia, Spain) and serves as a test bench of diverse technical solutions and configurations for future NEXT detectors. As stated in Section 1.2, the compression ratio of zero suppression in NEXT-NEW was around 70%. In NEXT-DEMO++, the SiPM sensor is the Hamamatsu S13372-1350TE (the same that will be mounted in NEXT-100) instead of SensL (series-C). To obtain similar results with both detectors, the threshold in the test has been adjusted to a different ADC count value (7 instead of 10 in NEXT-NEW) because the Hamamatsu SiPM sensors are operated at a lower gain (70%). Table 5 summarizes the results obtained.

As it can be observed in Table 5, with the sensors that are going to be used in the NEXT-100 detector, the compression ratio of the zero-suppression module is 71.90%, very similar to the obtained during the NEXT-NEW operation. Adding to this configuration the compression module, the compression ratio rises to 78.44%, and an extra 7.38% is achieved. In addition, the compression ratio of the compression module is just 9.46% lower than that of the zero-suppression module. In both cases, the result is very similar to what is shown in Section 6.3, which has been obtained with NEXT-NEW data simulations. The compression module has only one Huffman tree available, and it is applied when the compression module is on. As shown in Section 6.3, the impact of using the same Huffman tree with ZS and non-ZS data is negligible.

Table 5 also shows that the ratio between events detected and lost reduces from 26% to 29% depending on the compression method applied, clearly helping to reduce the dead time of the data acquisition system.

The CPU time required to decode the data is affected by compression. Table 5 shows the decoding times for each run. The first run, without compression or zero suppression, can be taken as a benchmark to which the other configurations can be compared. Using only zero suppression since there is less data to be decoded, the time needed is reduced by more than 50%. The case where compression is used without zero suppression is slower than the raw case, which is to be expected given that the algorithm is more complex and requires more memory access to read the Huffman tree. Finally, combining both zero suppression and compression, the decoding time is slightly above that for only zero suppression, but overall, still about half the time needed with raw data. In the case of the PMTs, a similar effect is to be expected, but despite the ~30% increase in processing time, the current computing equipment can handle the load.

## 8. Summary and Conclusions

This article has presented the compression algorithm implemented for the acquisition system of the NEXT-100. Different solutions have been studied for the type of sensors used in NEXT detectors, choosing the best solution in terms of compression efficiency, simplicity of the algorithm and minimum amount of hardware resources. One of the key features of the compression module presented, in addition to its high compression ratio, is that it is adaptable to different types of sensors, as happens in NEXT detectors. This facilitates the implementation and decoding process of the data.

An innovative compression method for SiPM sensors, merging lossy and lossless compression algorithms, has also been presented, with relevant results. Merging both types of compression methods allows the compression ratio to increase by around 7% without compromising the signal information in the NEXT-DEMO++ tests. This compression allows, with less demanding zero-suppression configurations, to reach very high compression ratios.

The adopted solution has shown very good results in terms of compression ratio, around 80% for both planes. The high compression ratio will help to avoid excess data throughput, minimizing the dead time of the NEXT-100 detector. Moreover, it will allow the data acquisition system to remain under the maximum throughput initially proposed of the system, avoiding scaling up the data acquisition system, which implies more complexity and cost.

NEXT future detectors will be more demanding in terms of the number of sensors and data buffering. This implies a higher throughput requirement for the acquisition system. The modules presented, implemented in this case in hardware or in the processing cores of future devices, will help to contain it.

## Figures and Tables

**Figure 1 sensors-22-05197-f001:**
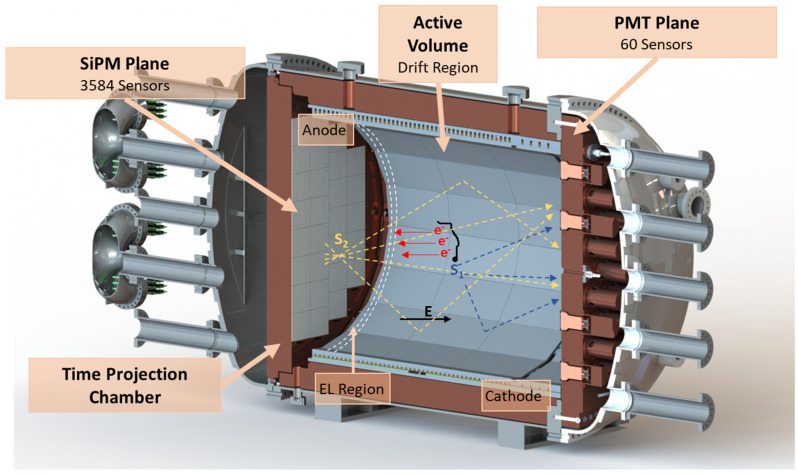
Schematic view of the NEXT-100 Detector. In the active volume of the TPC: drawing with the principle of operation of the detector.

**Figure 2 sensors-22-05197-f002:**
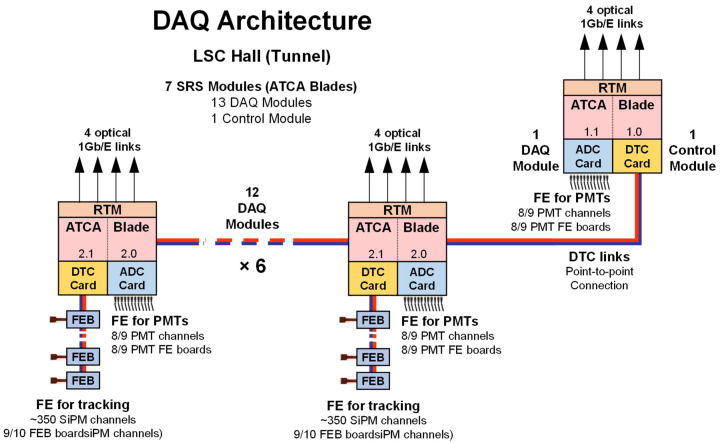
NEXT-100 Data Acquisition and Front-End (FE) hardware architecture.

**Figure 3 sensors-22-05197-f003:**
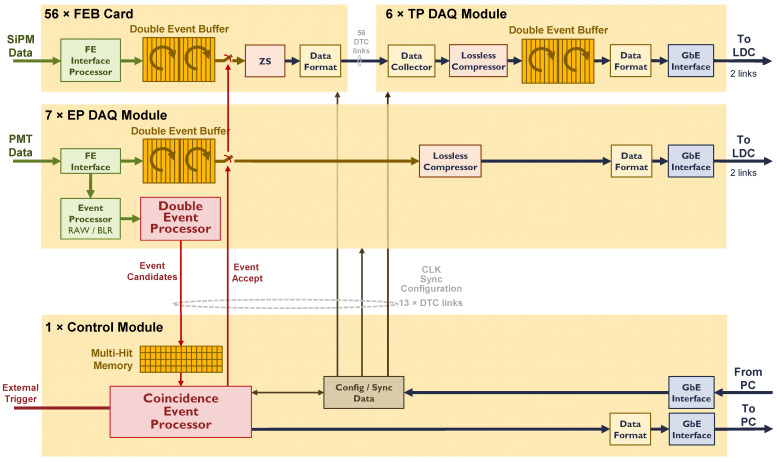
NEXT-100 data acquisition and event detection scheme.

**Figure 4 sensors-22-05197-f004:**
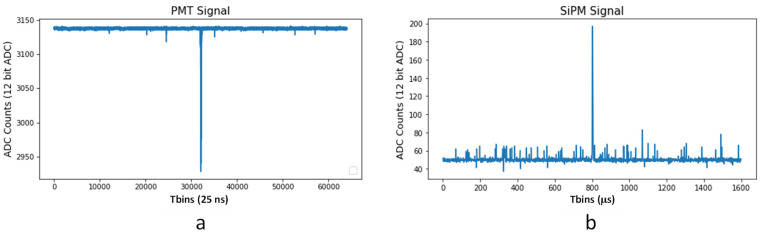
NEXT-White PMT and SiPM events (1.6 ms buffer): (**a**) PMT sensor signal; (**b**) SiPM sensor signal.

**Figure 5 sensors-22-05197-f005:**
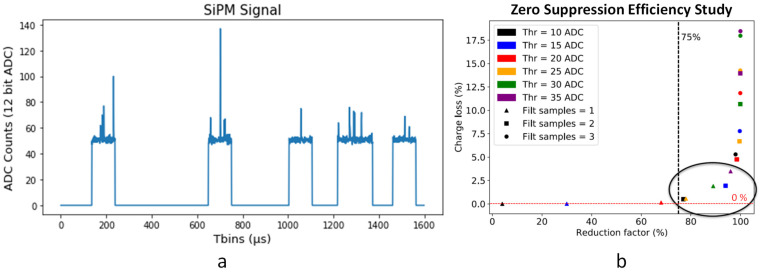
Zero Suppression in the NEXT-White TP: (**a**) Typical zero-suppressed signal from a SiPM sensor; (**b**) Charge loss vs. Reduction factor (Compression) in the zero-suppressed events of the detector with different set of parameters (pre- and post-samples fixed to 50 µs in all cases). This study has been done using the Dual Mode implemented in the DAQ (this mode allows the DAQ to send both sets of data, ZS and the raw versions of the events).

**Figure 6 sensors-22-05197-f006:**
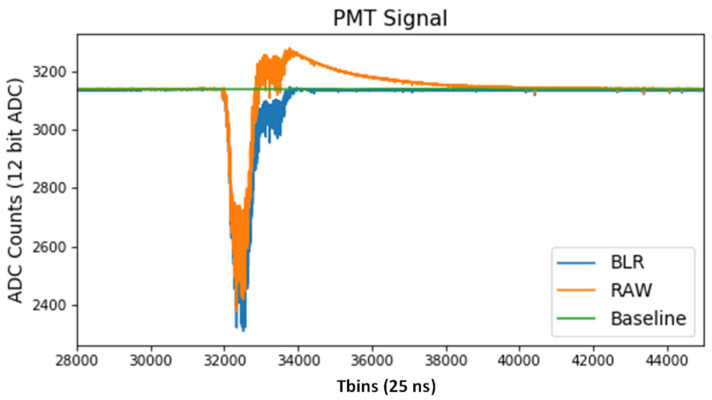
RAW and Baseline Restored (BLR) PMT signals.

**Figure 7 sensors-22-05197-f007:**
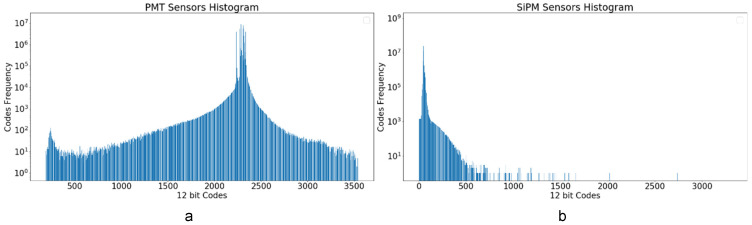
Sensor Histograms: (**a**) EP Histogram from run 6203 (^83m^Kr and ^232^Th sources, 12 PMTs, 346 events); (**b**) TP Histogram from run 8087 (1792 SiPMs, 307 events). In both cases, the sensor data are digitized with a 12-bit ADC.

**Figure 8 sensors-22-05197-f008:**
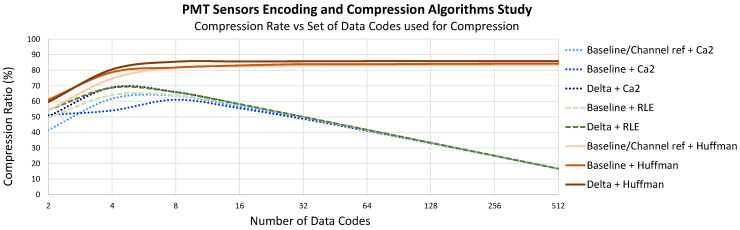
EP sensor compression ratio versus different numbers of 12-bit data codes used for compression over different data encoding applied.

**Figure 9 sensors-22-05197-f009:**
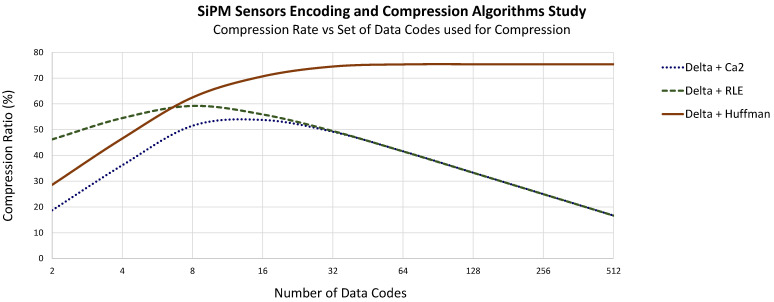
TP sensor compression ratio versus different numbers of 12-bit data codes used for compression over different data encoding applied.

**Figure 10 sensors-22-05197-f010:**
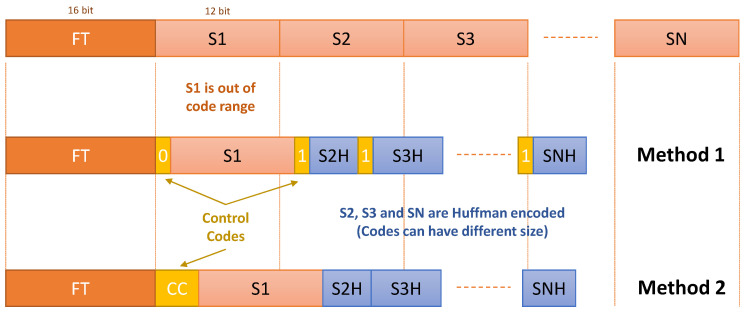
Control codification proposed methods.

**Figure 11 sensors-22-05197-f011:**
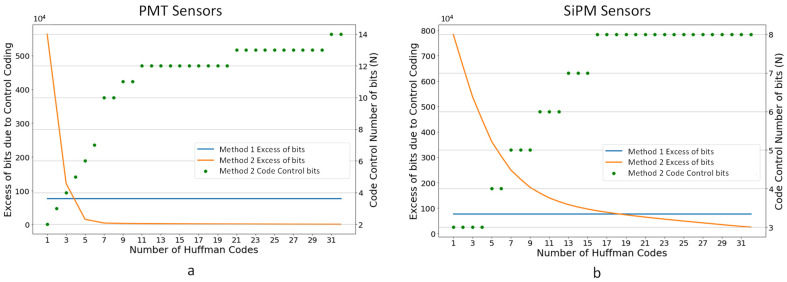
Excess of bits due to control coding versus the number of Huffman codes used to compress the data stream: (**a**) EP sensor signal from run 7175 (^83m^Kr source and low background, 12 PMTs, 1 event); (**b**) TP sensor signal from run 8087 (^83m^Kr source and low background, 1792 SiPM sensors and 1 event).

**Figure 12 sensors-22-05197-f012:**
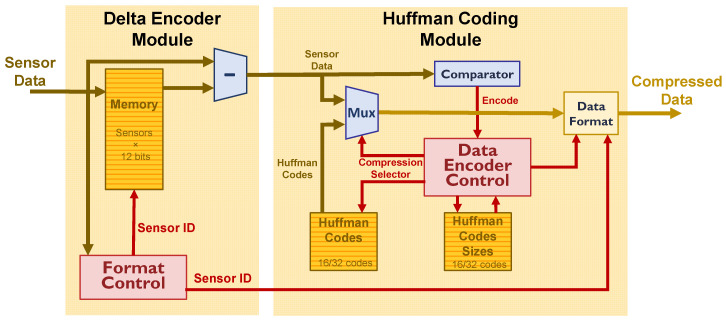
Compression Module Scheme.

**Figure 13 sensors-22-05197-f013:**
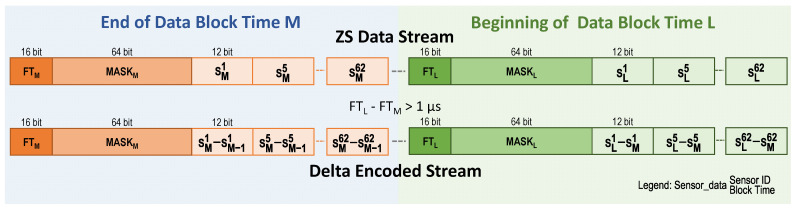
Delta encoding for the TP sensors when applied to zero-suppressed data. Fine Timer (FT) and sensor Mask (MASK) are present for each event time bin if at least one sensor has data. It is assumed that in both data blocks, data from sensors 1, 5 and 62 are present.

**Figure 14 sensors-22-05197-f014:**
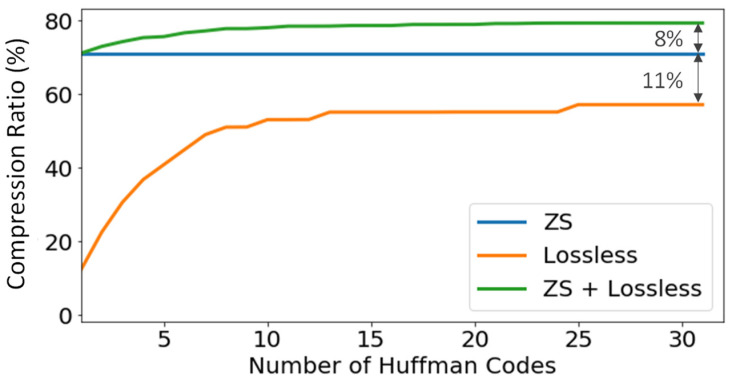
TP sensor compression ratio comparison. ZS and ZS + lossless compression data from run 8720 (611 events) and only lossless compression data from run 8087 (109 events), both runs with ^83m^Kr source and low background, 1792 SiPM sensors.

**Figure 15 sensors-22-05197-f015:**
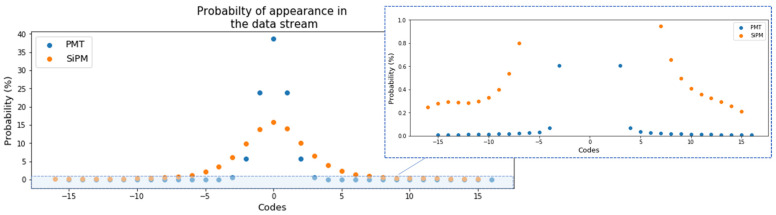
Left: probability of appearance in the data stream of a set of codes with a range of 32 12-bit values from RUN 8087 (109 events), both with ^83m^Kr source and low background. Right: a detail of the codes with probability under 1%.

**Figure 16 sensors-22-05197-f016:**
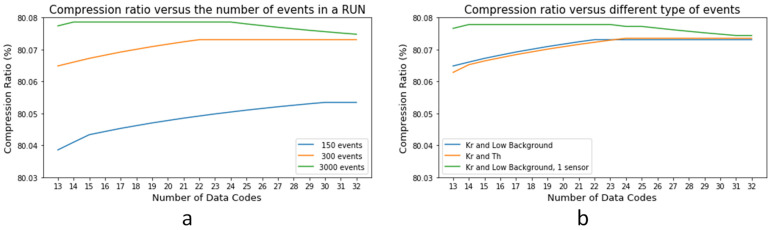
Variation in the compression ratio for run 6323 (3000 events) due to: (**a**) the number of events to set the Huffman tree and (**b**) the type of data source. Huffman codes obtained with 300 events for run 6323 (^83m^Kr source and low background) and run 6203 (^83m^Kr source and ^228^Th).

**Figure 17 sensors-22-05197-f017:**
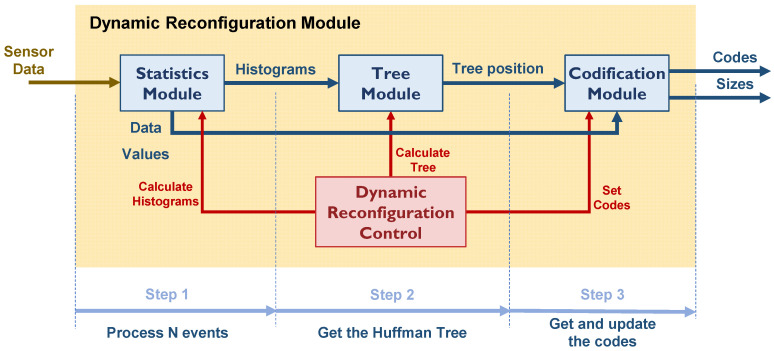
Huffman Codes Dynamic Reconfiguration Module scheme.

**Table 1 sensors-22-05197-t001:** Compression ratio versus code control codification methods. Method 2b stays for Method 2 with code control codification of 4 bits and Huffman codes shifted from this one. 32 Huffman codes. EP sensor signal from run 7175 (12 PMTs, 312 events). TP sensors from run 8167 and from run 8720 (1792 SiPM sensors, 109 and 611 events, respectively).

	Compression Ratio (%)		
	EP DAQ Module	TP DAQ Module	
	Raw	Raw	ZS
Method 1	80.07	57.14	80.12
Method 2	86.23	63.27	80.85
Method 2b	85.26	63.27	80.85

**Table 2 sensors-22-05197-t002:** Hardware resources needed for the Compression Module implemented in Xilinx Virtex-6 FPGAs (XC6VLX240T-1ff1156) in the EP and TP DAQ Modules.

	EP DAQ Module		TP DAQ Module	
	Compression Module	Total	Compression Module	Total
Slice Registers	0.15%	12%	0.11%	10%
Slice LUTs	0.57%	36%	0.44%	20%
Occupied Slices	0.88%	46%	0.79%	45%
RAM36E	0%	24%	0%	20%
RAM18E	0%	5%	0.12%	5%
DSP48E	0.26%	40%	0.13%	0.13%

**Table 3 sensors-22-05197-t003:** Hardware resources needed for the Dynamic Reconfiguration Module implemented in a Xilinx Virtex-6 FPGAs (XC6VLX240T-1ff1156) in comparison with the resources used in the EP and TP DAQ modules.

	Dynamic Reconfiguration Module		DAQ Modules
	Used	Utilization	Utilization
Slice Registers	3527	3.79%	12/10%
Slice LUTs	3609	7.75%	36/20%
Occupied Slices	1158	9.95%	46/45%
RAM18E	3	0.96%	5/5%

**Table 4 sensors-22-05197-t004:** Two run statistics in NEXT-NEW with general configuration: 12 PMT sensors, event rate around 20 Hz, 10,000 events, 1.6 ms buffer, searching for ^83m^Kr and low background events, at least 2 PMT hits in a Coincidence Window of 1.2 μs.

	Compression	Data Size (MBytes)	Compression Ratio (%)
Run 7299	OFF	18,601.53	0
Run 7298	ON	3283.47	82.35

**Table 5 sensors-22-05197-t005:** Three run statistics with general configuration: 256 SiPM sensors, event rate around 70 Hz, 10,000 events, 1.6 ms buffer, searching for ^83m^Kr, at least 2 PMT hits in a Coincidence Window of 1.2 μs, ZS set with 7 ADC counts over the threshold, 50 pre- and post-samples.

	ZS	Compression	Data Size (MBytes)	Compression Ratio (%)	Ratio Events/Losts	Processing Time (s)
Run 11233	OFF	OFF	6109.31	0	1.80	270.96
Run 11234	ON	OFF	2140.18	71.06	1.27	132.85
Run 11235	ON	ON	1316.90	78.44	1.20	151.83
Run 11236	OFF	ON	2346.04	61.60	1.34	347.09

## Data Availability

Data collected through research presented in the paper are available on request from the corresponding authors.

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
