# Peer review of "Data Compression in the NEXT-100 Data Acquisition System"

_sensors, 2022, doi:10.3390/s22145197_

Round 1
Reviewer 1 Report
Article ID: sensors- 1807759
Data Compression in the NEXT-100 data acquisition system
The paper concentrates on the evaluation of lossless data compression methods suitable for the NEXT-100 detector taking into consideration the limited and specific data processing resources available for data compression mainly based on FPGA devices. The main aim of research is to reduce data throughput, minimize detector dead time and keep event rate close to 50 Hz.
The paper presents a study of different techniques for lossless compression and the alternatives in the codification process. The final measured compression ratio was around 80%.
This is a good quality well-structured paper presenting valid research.
There are a few minor and major problems in the paper that need to be clarified and corrected before the publication. Some of additional issues are summarised below (MA=major, MI=minor, x/z means page/line):
MI 4/168: add space between unit and number: 1MHz -> 1 MHz
MI 3/89: always explain the acronym or shortcut when used for the first time, e.g TPB, DTC (4/166), HDMI, SRS, NIM, FE, ZS (fig. 2)
This paper is suitable for MDPI publication, however a few minor corrections are needed before the publication.
Author Response
Rev 1
MI 4/168: add space between unit and number: 1MHz -> 1 MHz CORRECTED
MI 3/89: always explain the acronym or shortcut when used for the first time, e.g TPB, DTC - CORRECTED
(4/166), HDMI, SRS, NIM, FE, ZS (fig. 2) CORRECTED, and some more (FEB, GbE,...).
Rev 2
In Sección 6.3, line 553, It has been included the following sentence to clarify the test:
"In this test, different Huffman encoding tree have been applied for Lossless compression and ZS + Lossless compression. Anyway, with the same Huffman trees applied to both cases (the one obtained for non-ZS data), the compression ratio differs by 0.00015 %. This means that, even being the Huffman trees obtained slightly different, ZS does not have a significant impact on the code frequency."
And in line 679:
"The compression module has only one Huffman tree available. As it has been shown in Section 6.3, the impact of using the same Huffman tree with ZS and non-ZS data is negligible.
Reviewer 2 Report
The authors present the data compression techniques used in the data acquisition system for the NEXT-100 detector, a neutrinoless double beta decay experiment in Xenon.
The experimental setup is well described and the high demands on energy resolution are well motivated. Extensive calibration is needed which in turn leads to higher event and data rates compared to the previous experiment NEXT-White. Although the DAQ system has been scaled appropriately, the maximum system throughput is below the needs of the calibration procedure. Improving data compression hence helps reaching those needs without increasing the DAQ system or running the calibration over a longer time period.
The studies of real-time data compression implemented in hardware are described in great detail and the results are well presented. I have only a single question for the authors related to Figure 14. In the scenario of Huffman encoding with and without zero-suppression -> is the exact same Huffman encoding tree applied? Or does the zero-suppression not have a significant impact on the code frequency? It could be helpful to explicitly state this in the document.
Author Response

(The authors gave the same response as above.)
